# Structural modeling for Oxford histological classifications of immunoglobulin A nephropathy

**Kensuke Joh**[1]*, **Takashi Nakazato**[2], **Akinori Hashiguchi**[3], **Akira Shimizu**[4], **Ritsuko Katafuchi**[5], **Hideo Okonogi**[6], **Kentaro Koike**[7], **Keita Hirano**[7], **Nobuo Tsuboi**[7], **Tetsuya Kawamura**[7], **Takashi Yokoo**[7], **Ichiei Narita**[8], **Yusuke Suzuki**[9]

1 Department of Pathology, The Jikei University Graduate School of Medicine, Tokyo, Japan, 2 Department of Medical Information Management, NHO Chiba Medical Center, Chiba, Japan, 3 Department of Pathology, Keio University School of Medicine, Tokyo, Japan, 4 Department of Pathology, Nihon Medical School, Tokyo, Japan, 5 Kidney Unit, National Hospital Organization Fukuoka-Higashi Medical Center, Fukuoka, Japan, 6 Division of Nephrology and Hypertension, Atsugi City Hospital, Kanagawa-ken, Japan, 7 Division of Nephrology and Hypertension, Department of Internal Medicine, The Jikei University School of Medicine, Tokyo, Japan, 8 Division of Clinical Nephrology and Rheumatology, Niigata University Graduate School of Medical and Dental Sciences, Niigata, Japan, 9 Division of Nephrology, Juntendo University Faculty of Medicine, Tokyo, Japan

* johken@med.tohoku.ac.jp

**Data Availability Statement:** All relevant data are within the paper and its Supporting information files.

## Abstract

In immunoglobulin A nephropathy (IgAN), Cox regression analysis can select independent prognostic variables for renal functional decline (RFD). However, the correlation of the selected histological variables with clinical and/or treatment variables is unknown, thereby making histology-based treatment decisions unreliable. We prospectively followed 946 Japanese patients with IgAN for a median of 66 mo. and applied structural equation modeling (SEM) to identify direct and indirect effects of histological variables on RFD as a regression line of estimated glomerular filtration rate (eGFR) via clinical variables including amount of proteinuria, eGFR, mean arterial pressure (MAP) at biopsy, and treatment variables such as steroid therapy with/without tonsillectomy (ST) and renin–angiotensin system blocker (RASB). Multi-layered correlations between the variables and RFD were identified by multivariate linear regression analysis and the model's goodness of fit was confirmed. Only tubular atrophy/interstitial fibrosis (T) had an accelerative direct effect on RFD, while endocapillary hypercellularity and active crescent (C) had an attenuating indirect effect via ST. Segmental sclerosis (S) had an attenuating indirect effect via eGFR and mesangial hypercellularity (M) had accelerative indirect effect for RFD via proteinuria. Moreover, M and C had accelerative indirect effect via proteinuria, which can be controlled by ST. However, both T and S had additional indirect accelerative effects via eGFR or MAP at biopsy, which cannot be controlled by ST. SEM identified a systemic path links between histological variables and RFD via dependent clinical and/or treatment variables. These findings lead to clinically applicable novel methodologies that can contribute to predict treatment outcomes using the Oxford classifications.

**Funding:** This study was supported in part by a Grant-in-Aid for Progressive Renal Diseases Research, Research on Rare and Intractable Disease, from the Ministry of Health, Labour and Welfare of Japan (Grant Number:20FC1045). This research was supported by AMED under Grant Number JP19ek0109261. The funders had no role in study design, data collection and analysis, decision to publish, or preparation of the manuscript.

**Competing interests:** The authors have declared that no competing interests exist.

## Introduction

The international histological classification for IgA nephropathy (IgAN) known as the Oxford classification was developed by defining and selecting relevant pathological variables for renal functional decline (RFD). This classification utilizes five pathological variables assessed individually: mesangial hypercellularity (M), endocapillary hypercellularity (E), segmental glomerulosclerosis (S), tubular atrophy/interstitial fibrosis (T), and cellular or fibrocellular (active) crescent (C) [1–3]. Almost all previous studies used traditional Cox regression analysis, which selects independent prognostic variables to propose high-risk patients. However, the correlation of the selected histological variables with clinical and/or treatment variables is unknown, and this is important information when choosing therapy. As a consequence, high-risk patients, irrespective of either active or chronic histological variables, equally receive immunosuppressive treatment at the time of biopsy [4, 5]. Due to a lack of sufficient evidence for histology-based decision making, the Kidney Disease: Improving Global Outcomes (KDIGO) guidelines recommend the choice of immunosuppressant therapy be made, not based on histology, but mostly based on clinical features indicating more than 1 g/d proteinuria around the time of biopsy and during the two years after biopsy [6]. Therefore, risk stratification and treatment decisions currently still rely on inaccurate categorization of these risk factors.

Recently, a new International Risk-Prediction Tool for IgAN has been proposed as a more accurate tool to predict disease progression based on both histological and clinical risk factors, including treatment choice [7–10]. This is a personalized prediction model using an equation composed not only of clinical variables including urine protein excretion (UPE), estimated glomerular filtration rate (eGFR), and mean arterial pressure (MAP) at biopsy and treatment choice such as steroid therapy (ST) and renin–angiotensin system blocker (RASB) but also the histological variables MEST. This personalized equation predicts the probability of developing 50% decrease in eGFR or end-stage renal disease in 5 years [8]. However, the formula is composed of evenly evaluated histological, clinical, and treatment variables as a simple summation in the exponential function [8, 11]. Therefore, it is still not known how each histological variable will respond to treatment choice after renal biopsy.

RFD is associated not only directly with histological, clinical, and/or treatment variables, but also indirectly and unequally with clinical and/or treatment variables. Structural equation modeling (SEM) is a method which can estimate these complex interactions by adjusting for measurement errors in dependent variables using the error term "ε," reducing bias in correlation estimates [12, 13].

In this prospective study, we therefore aimed to use SEM with multivariate linear regression analysis to find structural paths of correlation between each histological variable and slope as a regression line of eGFR (SLOPE) via clinical variables and treatment choice. The appropriately fitting model, including direct and indirect effects on RFD of Oxford classification variables via clinical and/or treatment variables, can then be implemented clinically.

## Materials and methods

### Objectives

This was a prospective clinical study. Patients with IgAN were recruited and their clinical data and renal biopsy materials collected at 44 kidney centers across Japan. This clinical research project was produced by the ethical committee of The Research Group on Progressive Renal Diseases organized by the Ministry of Health, Labour and Welfare in Japan. The study protocol was in accordance with the standards of the ethics committee at each center, and each

patient consented to participate after being informed of the purpose and procedure of the study.

Patients with IgAN registered between April 2005 and August 2015 in the Japan IgA Nephropathy Prospective Cohort Study (JIGACS), which is a prospective observational study conducted at facilities throughout Japan, were included in the study. The protocol was approved by the Jikei University School of Medicine's Institutional Review Board on Human Research, which acted as the main ethics committee for this study (No. 16–174 [4402]). Written consent was obtained from each patient and the date of consent acquisition was recorded in the designated column of the consent form. If the patient was below the age of 18 years, written consent was obtained from his/her guardian. Of 1130 patients, we excluded those with more than one of the following variables missing: MESTC score, baseline UPE, eGFR, or MAP, and SLOPE measurements, or if they had no follow-up data after renal biopsy. Secondary cases that showed mesangial IgA deposits, although with a predominant combined disease such as diabetes mellitus, were excluded. After exclusions, 946 patients were included in the study.

## Clinical data

Normally distributed variables, as determined using the Shapiro–Wilk test, non-parametric variables and categorical variables were expressed as mean ± standard deviation, as median and range, and as frequency and percentage, respectively. The data were analyzed using SPSS version 24 (IBM Corp., USA).

Clinical variables, which were collected within 1 mo of biopsy and during follow-up, were as follows. MAP was defined as diastolic pressure plus a third of the pulse pressure. The eGFR was calculated as per the Japanese-based equation: eGFR (ml/min/1.73 m$^2$) = 194 × serum creatinine (sCr)$^{-1.094}$ × age$^{-0.287}$ (if female, ×0.739) [14]. In patients who were younger than 20 years, the eGFR was calculated incorporating polynomial formulas for sCr and body length [15]. The Oxford study excluded initial GFR <30 ml/min and initial proteinuria <0.5 mg/d, but no such exclusion criteria were applied in this study. Treatments taken within a year after biopsy were recognized as selected treatments: immunosuppressive treatment including ST with/without tonsillectomy (reported as an intended treatment regardless of the type or duration of therapy) or RASB indicating any exposure to angiotensin-converting enzyme inhibitor, angiotensin receptor blockers, or both. If the patients had already received RASB at the time of renal biopsy, this was also recorded as an initial treatment. Weight, height, sCr, and amount of proteinuria (g/d) were also recorded.

## Outcomes

In patients where more than two eGFR measurements were available, SLOPE was calculated as a regression line and used as an outcome, as an indicator of RFD.

## Pathological data

All cases were proven to be IgAN by biopsy. IgAN was defined by dominant mesangial depositions of IgA and its presence in more than 10 glomeruli. Pathological variables used in the present study included M0, M1, E0, E1, S0, S1, T0, T1, or T2, and C0, C1, or C2, which were defined according to the Oxford classification [1–3]. Briefly, M0 and M1 indicated the percentages of glomeruli with a mesangial hypercellularity of 0%–50% and <50%, respectively according to a simplification of the mesangial hypercellularity score as shown in the Oxford study [1, 2]. S0 or S1 and E0 or E1 indicated the absence or presence of S and E, respectively. T0, T1, and T2 indicated interstitial fibrosis at 0%–25%, 26%–50%, and >50%, respectively. C0, C1, and C2 indicated the percentage of glomeruli with either a cellular or fibrocellular crescent at

0%, 1%–24%, and ≥25%, respectively. Renal biopsies were scored for pathological variables according to the Oxford study by five renal pathologists (KJ, AH, AS, SH, RK) blinded to the clinical data. When disagreement occurred between observers, scoring was repeated by all five observers to reach a consensus. The agreement rate was good or moderate in our previous study; the intraclass correlation coefficient of M, E, S, T1/T2, and C1/C2 were 0.54, 0.57, 0.64, 0.72, and 0.57, respectively, among the five pathologists [16].

## SEM

SEM model building and estimation was done using STATA/SE version 15 (Light Stone, USA). A two-sided $P < 0.05$ was considered significant.

SEM included analysis of the direct paths between histological variables and SLOPE and indirect paths between histological variables and SLOPE via clinical factors and treatment choice, and linear regression analysis was used to find statistically significant direct and indirect effects. Linear regression was used instead of Cox regression to find indices of the model's statistical fit, as goodness of fit could not be applied for the survival analysis model based on Cox multivariate analysis.

The candidate variables were selected according to a previous study by Barbour et al. [8] We first created a hypothetical model consisting of all histological, clinical, and treatment variables and SLOPE as a marker of RFD (Fig 1). Considering normality, clinical variables were modified as follows: The baseline UPE0, baseline square root eGFR0 (SReGFR0), and MAP at biopsy were centralized as each patient's UPE0c, SReGFR0c, and MAPc, which was the value subtracted by an average of 946 patients in each category. Direct paths are indicated by arrows from each independent histological variable (M, E, S, T, and C), clinical variable (centralized UPE [UPEc0], centralized square root eGFR [SReGFR0c], or centralized MAP dichotomized with negative MAPc as MAPc0 and positive MAPc as MAPc1 [MAPc01]), and treatment

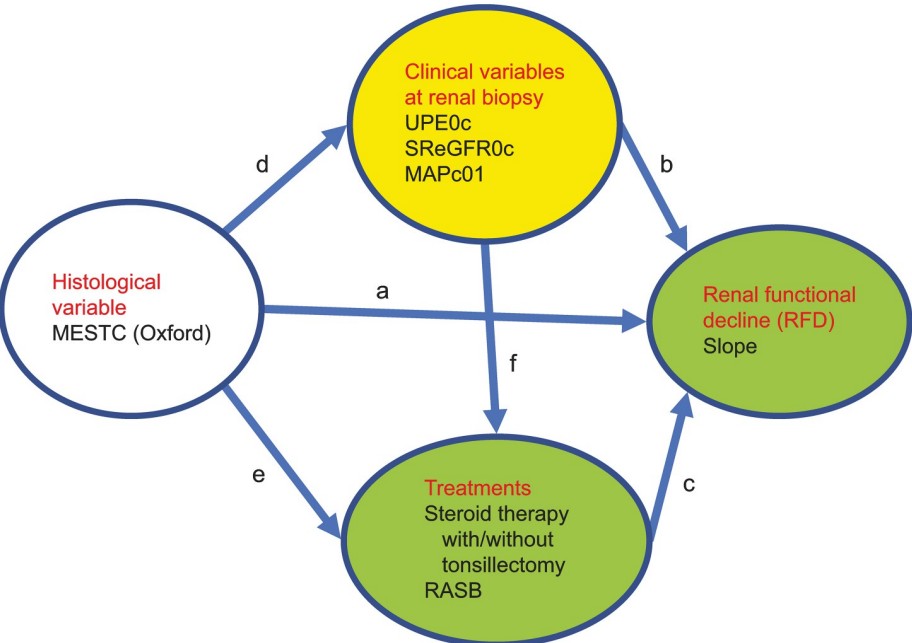

**Fig 1. Hypothetical model incorporating all histological, clinical, and treatment variables and the slope of change in estimated glomerular filtration rate (SLOPE), as a marker of renal functional decline (RFD).**

variable (ST and RASB) after renal biopsy leading to SLOPE as an endpoint (Fig 1a–1c). Indirect paths are indicated by arrows from each independent histological variable to each mediating variable (clinical and treatment; Fig 1d and 1e) and from each dependent clinical variable to each dependent treatment variable (Fig 1f). As MAP and eGFR can affect the amount of proteinuria [6], the correlations between UPE0c and MAPc01 or SReGFR0c were added as clinical variables. The correlations of a–f were estimated using a multiple linear regression model. We reported standardized path coefficients (SC), $P$ values, and confidence intervals (CI). In addition, error terms ($\varepsilon 1$–$\varepsilon 6$), which account for measurement error, were used in addition to latent clinical variables and SLOPE.

Direct paths are indicated as arrows from each independent histological variable (M, E, S, T, or C), clinical variable (UPEc0, SReGFR0c, and MAPc01), and treatment variable (ST with/without tonsillectomy and renin–angiotensin system blocker [RASB]) after renal biopsy leading to SLOPE (Fig 1a–1c). Indirect paths are indicated as each histological variable pointing to each mediating variable (clinical and treatment, Fig 1d and 1e) and pointing to each dependent treatment variable from each dependent clinical variable (Fig 1f).

These calculations using the hypothetical model identified significant correlations among histological, clinical, and treatment variables and SLOPE, allowing a new model to be drawn. Only paths with correlations of $P < 0.1$ were drawn as arrows in this revised model. To determine any elimination bias, it was checked whether link paths with correlations of $P < 0.1$ were the same as those with correlations of $P < 0.05$. An appropriately fitting model was proven using population error (root mean square error of approximation [RMSEA] $< 0.05$ with 90% CI), baseline comparison (comparative fit index [CFI] $> 0.90$), and size of residuals (standardized root mean square residual [SRMSR] $< 0.05$) [17–19]. The sample size of 946 observations for the present analysis consisting of 19 variables including M0, M1, E0, E1, S0, S1, T0, T1 or T2, C0, C1 or C2, UPE0c, SReGFR0c, MAPc0, MAPc1, ST0, ST1, RASB0, RASB1, and SLOPE was appropriate because an ideal sample size-to-parameters ratio would be 20:1, indicating that the minimum sample size should be $19 \times 20$ [20].

## Results

### Clinical profile

Clinical profiles of 946 Japanese with IgAN are shown in Table 1. Patients had a median age of 37.1 years old (2.8–87.5 years old) and were followed up for a median of 66 mo (1–174 mo). There was an equal distribution of males (49%) and females (51%) indicating no gender-based skewing of the results. MAP at renal biopsy was $90.0 \pm 13.7$ mmHg and dichotomized and centralized MAPc1 was 45%. eGFR0 was $75.6 \pm 28.9$ ml/min and UPE0 was $1.1 \pm 2.2$ mg/dl. Body mass index (BMI) was $22.1 \pm 3.8$. The SLOPE was $-0.10 \pm 0.51$ ml/min/1.73 m$^2$/y.

### Treatment

Table 1 shows an overview of treatment choices for the study participants. ST was used to treat 64%, which included steroid pulse therapy in 20%, steroid pulse with tonsillectomy in 38%, oral steroid therapy in 5%, and oral steroids with tonsillectomy in 1%. RAS blockade was used to treat 57%. A small proportion (5%) was treated only with tonsillectomy, and 34% were not treated at all.

### Pathological profile

Histological results from renal biopsy samples are shown in Table 1. The proportion of the patients with M1, E1, S1, T1, T2, C1, and C2 was 29%, 35%, 74%, 18%, 4%, 38%, and 1%, respectively.

**Table 1. Clinical, treatment, and histological profiles of the IgAN patients in this study.**

| Clinical profile | | Histological profile | |
|---|---|---|---|
| | | | n (%) |
| Cohort | 946 | M | |
| Male, n (%) | 463 (49%) | M0 | 671 (71%) |
| Female, n (%) | 483 (51%) | M1 | 275 (29%) |
| Age | 37.1 (2.8–87.5) | E | |
| Ethnicity | Japanese | E0 | 611 (65%) |
| MAP at biopsy (mmHg) | 90.0 ± 13.7 | E1 | 335 (35%) |
| MAPc01, n (%) | | S | |
| Yes | 430 (45%) | S0 | 247 (26%) |
| No | 516 (55%) | S1 | 699 (74%) |
| eGFR (ml/min/1.73 m$^2$) | 75.6 ± 28.9 | T | |
| UPE0 (at biopsy) | 1.1 ± 2.2 | T0 | 739 (78%) |
| BMI | 22.1 ± 3.8 | T1 | 170 (18%) |
| eGFR slope (ml/min/1.73m$^2$/y) | −0.10 ± 0.51 | T2 | 37 (4%) |
| Period of follow-up (mo) | 66 (1–174) | C | |
| | | C0 | 580 (61%) |
| Treatment choice | n (%) | C1 | 357 (38%) |
| ST | | C2 | 9 (1%) |
| Yes | 605 (64%) | | |
| No | 341 (36%) | | |
| RASB | | | |
| Yes | 539 (57%) | | |
| No | 407 (43%) | | |

BMI: body mass index; eGFR: estimated glomerular filtration rate; MAP: mean arterial pressure; MAPc01: centralized dichotomized MAP; RASB: renin-angiotensin system blocker; ST: steroid therapy with/without tonsillectomy; UPE0: baseline urine protein excretion. M0 and M1 indicated the percentages of glomeruli with a mesangial hypercellularity of 0%–50% and <50%, respectively. S0 or S1 and E0 or E1 indicated the absence or presence of S and E, respectively. T0, T1, and T2 indicated interstitial fibrosis at 0%–25%, 26%–50%, and >50%, respectively. C0, C1, and C2 indicated the percentage of glomeruli with either a cellular or fibrocellular crescent at 0%, 1%–24%, and ≥25%, respectively.

## SEM

A hypothetical model was drawn as described in the Methods and tested with a multivariate linear regression model using SEM (Fig 1). After removing non-significant paths (*P* >0.1), SEM was performed again to find an appropriately fitting model. The significant paths were the same whether the significance threshold was set at *P* <0.05 or *P* <0.1 as shown in Fig 2. Therefore, no elimination bias was apparent. As indices of the models' statistical fit, RMSEA, CFI, and SRMSR were 0.05 as same as 0.05, 0.93 as more than 0.90, and 0.03 as less than 0.05, respectively. Therefore, these results indicate that the model was an appropriate fit according to a likelihood estimation of appropriate fit as shown in the Methods [17, 18]. An analyzing process was shown in a flow chart (S1 File).

Statistically significant paths between histological variables, clinical variables, or treatment variables and SLOPE are shown as arrows with standardized coefficients marked. T, besides ST, UPE0c, and SReGFR0c, showed direct correlations with SLOPE (P < 0.05) (blue arrows). All histological variables showed indirect effects on SLOPE via clinical variables or ST (red arrows).

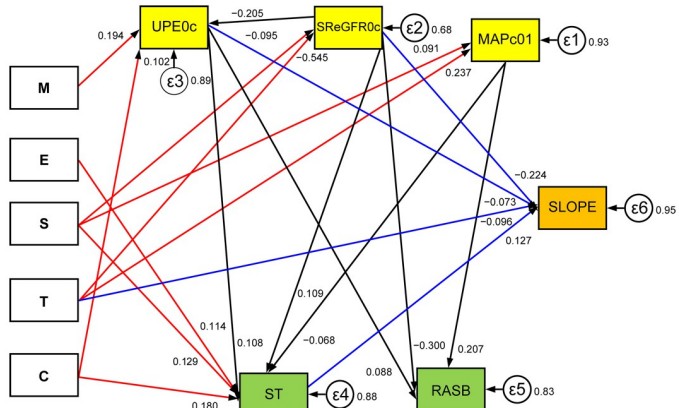

**Fig 2. SEM was performed to find an appropriate fitting model after removing non-significant paths ($P > 0.05$).**

Analyzing indirect contributors to SLOPE, ST, which correlated with S, E, C, UPE0c, SReGFR0c, and MAPc01, was correlated with SLOPE. RASB, which correlated with UPE0c, SReGFR0c, and MAPc01, was not correlated with SLOPE. The error terms ε1–ε6 of each of MAPc01, SReGFR0c, UPEc0, ST, RASB, and SLOPE were 0.93, 0.68, 0.89, 0.88, 0.83, and 0.95, respectively.

In the SEM analysis, statistically significant direct correlations with SLOPE ($P < 0.05$) were independent accelerative histological variable T, independent accelerative clinical variables UPE0c and SReGFR0c, and independent attenuating treatment variable ST. As well as the direct correlations with SLOPE, correlations between clinical and histological variables and between treatment and histological variables or clinical variables were calculated and indicated indirect correlations with SLOPE (Fig 2 and Table 2). UPE0c positively correlated with M and C and negatively correlated with SReGFR0c. MAPc01 positively correlated with S and T. SReGFR0c negatively correlated with S and T. ST negatively correlated with MAP01c and positively correlated with UPE0c, SReGFR0c, S, E, and C. RASB, although correlated with UPE0c, MAPc01, and SReGFR0c, was not an independent variable for SLOPE.

As shown in Table 3, direct and/or indirect effects of histological variables on SLOPE were calculated as total coefficients by combining the direct and indirect coefficients for each histological variable (Table 3). An analyzing process was shown in a flow chart (S1 File). The total coefficients were the sums of the direct and indirect coefficients, where each indirect coefficient of M, E, S, T, or C in Table 3 was the integration of each coefficient of the clinical and treatment variables in Table 2 according to the paths shown in Fig 2, which stood between histological variables and SLOPE (Fig 2). T showed an total accelerative effect composed of the direct accelerative coefficient and the indirect accelerative coefficients as via SReGFR0c and ST and via MAPc01 and ST. E showed an attenuating indirect effect via ST. C showed an attenuating indirect effect composed of an attenuating indirect effect via ST and attenuating indirect effect via UPE0c and ST. S showed an attenuating indirect effect via SReGFR0c and additionally an accelerative indirect effect via SReGFR0c and ST, as well as via MAPc01 and ST. M showed an accelerative effect composed of accelerative indirect effect via UPE0c and an attenuating indirect effect via UPE0c and ST (Table 3). In summary, only T showed an accelerative direct effect on RFD, whereas E and C were not independent variables for RFD, correlated significantly with ST, and showed attenuating effects on RFD via ST. S showed attenuating effects on RFD via SReGFR0c. Both C and M had additional accelerative effects via UPE0c, which can

**Table 2. Direct and indirect correlations between histological variables and change in estimated glomerular filtration rate (SLOPE) via clinical variables and treatment variables.**

|  | SC | SE | z | P>z | [95% Confidence Interval] | |
|---|---|---|---|---|---|---|
| **SLOPE** | | | | | | |
| UPE0c | −0.073 | 0.033 | −2.200 | 0.028 | −0.138 | −0.008 |
| SReGFR0c | −0.224 | 0.039 | −5.800 | <0.001 | −0.299 | −0.148 |
| ST | 0.127 | 0.032 | 3.99 | <0.001 | 0.065 | 0.19 |
| T | −0.095 | 0.038 | −2.500 | 0.013 | −0.170 | −0.020 |
| _cons | −0.318 | 0.056 | −5.630 | <0.001 | −0.429 | −0.207 |
| **UPE0c** | | | | | | |
| SReGFR0c | −0.205 | 0.031 | −6.660 | <0.001 | −0.266 | −0.145 |
| M | 0.194 | 0.031 | 6.26 | <0.001 | 0.133 | 0.254 |
| C | 0.102 | 0.031 | 3.31 | 0.001 | 0.042 | 0.162 |
| _cons | −0.205 | 0.042 | −4.870 | <0.001 | −0.287 | −0.122 |
| **MAPc01** | | | | | | |
| S | 0.091 | 0.032 | 2.84 | 0.004 | 0.028 | 0.152 |
| T | 0.237 | 0.03 | 7.79 | <0.001 | 0.177 | 0.297 |
| _cons | 0.637 | 0.066 | 9.61 | <0.001 | 0.507 | 0.766 |
| **SReGFR0c** | | | | | | |
| S | −0.095 | 0.027 | −3.52 | <0.001 | −0.148 | −0.042 |
| T | −0.545 | 0.021 | −25.740 | <0.001 | −0.587 | −0.504 |
| _cons | 0.448 | 0.051 | 8.79 | <0.001 | 0.349 | 0.548 |
| **ST** | | | | | | |
| UPE0c | 0.108 | 0.032 | 3.41 | 0.001 | 0.046 | 0.17 |
| MAPc01 | −0.069 | 0.033 | −2.100 | 0.036 | −0.133 | −0.004 |
| SReGFR0c | 0.109 | 0.034 | 3.24 | 0.001 | 0.043 | 0.175 |
| S | 0.127 | 0.032 | 3.97 | <0.001 | 0.064 | 0.19 |
| E | 0.114 | 0.033 | 3.41 | 0.001 | 0.048 | 0.18 |
| C | 0.180 | 0.034 | 5.24 | <0.001 | 0.112 | 0.247 |
| _cons | 0.956 | 0.076 | 12.53 | <0.001 | 0.807 | 1.106 |
| **RASB** | | | | | | |
| UPE0c | 0.088 | 0.03 | 2.88 | 0.004 | 0.028 | 0.147 |
| MAPc01 | 0.207 | 0.031 | 6.61 | 0 | 0.146 | 0.269 |
| SReGFR0c | −0.300 | 0.031 | −9.550 | 0 | −0.361 | −0.238 |
| _cons | 0.976 | 0.049 | 19.87 | 0 | 0.88 | 1.072 |

SC: standardized coefficient; SE: standard error; T: tubular atrophy/interstitial fibrosis; M: mesangial hypercellularity; C: active crescent; S: segmental glomerulosclerosis; E: endocapillary hypercellularity; ST: steroid therapy including tonsillectomy; RASB: renin-angiotensin system blocker; UPE0c: centralized base line urine protein excretion; SReGFR0c: centralized square root baseline eGFR; MAPc01: centralized dichotomized baseline mean arterial pressure; cons: constant.

be controlled by ST, thus changing the accelerative effect to attenuating effect. On the other hand, both T and S had additional indirect accelerative effects on RFD via SReGFR0c or via MAPc01, which could not be controlled by ST, thus continuing to exhibit the accelerative effect. The error terms ε1–ε6 of each MAPc01, SReGFR0c, UPEc0, ST, RASB, and SLOPE were 0.93, 0.68, 0.89, 0.88, 0.83, and 0.95, respectively (Fig 2).

## Discussion

In this prospective multicenter study involving 946 Japanese IgAN patients, we applied SEM to evaluate structural correlations associated with the change in eGFR as RFD (SLOPE).

**Table 3. Direct and indirect effects of Oxford histological variables on change in estimated glomerular filtration rate (SLOPE).**

| Histological variable | Direct coefficient | Indirect coefficient | Indirect via | Total coefficient |
|:---:|:---:|:---:|:---:|:---:|
| M |  | −0.014 | UPE0c | −0.011 |
| M |  | 0.003 | UPE0c, ST |  |
| E |  | 0.014 | ST | 0.014 |
| S |  | 0.021 | SReGFR0c | 0.019 |
| S |  | −0.001 | SReGFR0c, ST |  |
| S |  | −0.001 | MAPc01, ST |  |
| T | −0.095 |  |  | −0.105 |
| T |  | −0.008 | SReGFR0c, ST |  |
| T |  | −0.002 | MAPc01, ST |  |
| C |  | 0.023 | ST | 0.024 |
| C |  | 0.001 | UPE0c, ST |  |

M: mesangial hypercellularity; E: endocapillary hypercellularity; S: segmental glomerulosclerosis; T: tubular atrophy/interstitial fibrosis; C: active crescent; SReGFR0c: centralized square root baseline eGFR; MAPc01: centralized dichotomized baseline mean arterial pressure; UPE0c: centralized baseline urine protein excretion; ST: steroid therapy including tonsillectomy.

Consequently, this analysis showed systemic path links between SLOPE and histological variables via clinical and/or treatment variables.

We identified the contributors to SLOPE as T (T1 or T2, independent accelerative histological variable), SReGFR0c and UPE0c (independent accelerative clinical variables), and ST (independent attenuating treatment variable). RASB was not an independent variable for SLOPE. Further, the selected contributors as well as the independent variables for SLOPE were connected via dependent histological, clinical, and/or treatment variables. Therefore, we investigated the statistically significant structural correlations among the histological, clinical, and treatment variables and SLOPE.

Using the coefficients of the direct and indirect correlations, we calculated the total effect of each histological variable on SLOPE (Table 3). T was the only histological variable with a direct (accelerative) effect on SLOPE. It had additional indirect effects (accelerative) via SReGFR0c or MAPc01, which could not be controlled by ST, thus maintaining the accelerative effect. Both E and C attenuated SLOPE via ST. M showed an overall accelerative effect on SLOPE, thereby incorporating an accelerative effect via UPE0c. Further, both M and C had additional accelerative effects on SLOPE via UPE0c, which was controlled by ST, thereby transforming the accelerative effect to attenuating effect. S had an attenuating indirect effect via eGFR. Moreover, both T and S had indirect accelerative effects via eGFR0c or MAP, which could not be controlled by ST, thus continuing with the accelerating effect. If S0 developed to S1, SReGFR0c decreased (Table 2). This negative correlation meant that if eGFR at biopsy was high, SLOPE declined strongly, while if it was low, there was a weaker decline. This is also true in normal kidneys [21].

The above findings suggest that ST was chosen for patients with E1 and C1 or C2 but not with T1 or T2 and that it effectively attenuated decline in eGFR. This is in partial agreement with another Japanese study, which found that patients with E1, S1, or C1 treated with ST had significantly better prognosis than the non-treatment group [22]. Similarly, in two other studies, the presence of E was strongly associated with subsequent ST, and there was a higher rate of decline in renal function in patients who were not treated with immunosuppressants [1] or corticosteroids [23]. Our study also showed that C and M had additional accelerating effects

via UPE0c on SLOPE, which were controlled and thus led to the change from accelerative to attenuating effect by ST. Therefore, ST can be considered to be effective for the patients with C and M by diminishing the accelerative effect of UPE0c. Other studies have shown that patients had an increased risk of disease progression with extremely increasing C, even with immuno-suppression [5, 24, 25]. T and S had additional accelerative effects via SReGFR0 or MAPc01, which could not be controlled by ST. This is consistent with the previous findings, which reported that S (as well as M and T) had predictive prognostic value for RFD [1, 5, 23]. However, the prognostic role of S1 for RFD can be small, when early ST prevented a progression from active crescent as C1/C2 to S. M appears to be a sensitive histological variable, as it can also be a risk factor for RFD via acceleration of proteinuria, when proteinuria cannot be controlled by ST [5, 26, 27]. However, the original Oxford study selected S1 and T1/T2 but not M1 for RFD in a multivariate linear regression model adjusted for initial eGFR, MAP, and UPE0 [1, 28]. The VALIGA study suggested also M1 was a steroid-responsive variable [26, 29, 30]. These conflicting results illustrate the benefit of analyzing both the direct and indirect effects.

We were able to stratify the Oxford histological classifications from the viewpoint of treatment response in predicting future outcomes. Several validation studies of the Oxford classification using Cox analysis showed merely different independent histological, clinical, and treatment variables without suggesting their concrete clinical use, which may depend on differences in the cohorts [4, 5, 31, 32]. SEM can clarify these differences by analyzing the correlations between Oxford histological variables and RFD via clinical and treatment variables.

In conclusion, SEM identified a systemic path links between histological variables and RFD via dependent clinical and/or treatment variables. We focused not only on direct effect but also on indirect effect of histological variables on RFD, where direct effects of clinical variables on RFD, such as SReGFR0c, UPE0c and ST, were influenced by histological variables. To the best of our knowledge, there has been no such systematic research previously to propose the correlation among histological, clinical, and treatment variables in depth. These findings lead to clinically applicable novel methodologies that can contribute to predict treatment outcomes using the Oxford classifications and improve care for IgAN patients using personalized medicine.

As a limitation, this prospective study using SEM cannot predict the prognosis of renal function by Cox regression analysis because goodness of fit could not be applied for the survival analysis model. Obtaining a more apparent eGFR slope needs longer period of follow-up.

## Supporting information

**S1 File.**
(DOCX)

## Acknowledgments

We thank the study participants for their kind cooperation. The authors would like to thank Enago (www.enago.jp) for the English Language review.

## Author Contributions

**Conceptualization:** Takashi Nakazato, Takashi Yokoo, Yusuke Suzuki.

**Data curation:** Kensuke Joh, Akinori Hashiguchi, Akira Shimizu, Ritsuko Katafuchi, Hideo Okonogi, Kentaro Koike, Keita Hirano, Nobuo Tsuboi, Tetsuya Kawamura, Takashi Yokoo, Ichiei Narita.

**Formal analysis:** Takashi Nakazato, Akinori Hashiguchi, Akira Shimizu, Ritsuko Katafuchi, Hideo Okonogi, Kentaro Koike, Keita Hirano, Nobuo Tsuboi, Tetsuya Kawamura, Takashi Yokoo, Ichiei Narita, Yusuke Suzuki.

**Methodology:** Kensuke Joh, Takashi Nakazato, Takashi Yokoo, Ichiei Narita, Yusuke Suzuki.

**Validation:** Kensuke Joh.

**Visualization:** Kensuke Joh.

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
