## [Decision Letter · Decision Letter 0]

9 Jun 2022

PONE-D-22-13403Structural modeling for Oxford histological classifications of immunoglobulin A nephropathyPLOS ONE

Dear Dr. Joh,

Thank you for submitting your manuscript to PLOS ONE. After careful consideration, we feel that it has merit but does not fully meet PLOS ONE’s publication criteria as it currently stands. Therefore, we invite you to submit a revised version of the manuscript that addresses the points raised during the review process. Please submit your revised manuscript by Jul 24 2022 11:59PM. If you will need more time than this to complete your revisions, please reply to this message or contact the journal office at plosone@plos.org. Please include the following items when submitting your revised manuscript:A rebuttal letter that responds to each point raised by the academic editor and reviewer(s). You should upload this letter as a separate file labeled 'Response to Reviewers'.A marked-up copy of your manuscript that highlights changes made to the original version. You should upload this as a separate file labeled 'Revised Manuscript with Track Changes'.An unmarked version of your revised paper without tracked changes. You should upload this as a separate file labeled 'Manuscript'.

We look forward to receiving your revised manuscript.

Kind regards,

Zhanjun Jia

Academic Editor

PLOS ONE

Journal Requirements:

"This study was supported in part by a Grant-in-Aid for Progressive Renal Diseases Research, Research on Rare and Intractable Disease, from the Ministry of Health, Labour and Welfare of Japan. This research was supported by AMED under Grant Number JP19ek0109261."

Additional Editor Comments:

Some comments from the experts need to be well addressed.

Reviewers' comments:

Reviewer's Responses to Questions

**Comments to the Author**

1. Is the manuscript technically sound, and do the data support the conclusions?

Reviewer #1: Yes

Reviewer #2: Partly

Reviewer #3: Partly

2. Has the statistical analysis been performed appropriately and rigorously? 

Reviewer #1: Yes

Reviewer #2: N/A

Reviewer #3: Yes

3. Have the authors made all data underlying the findings in their manuscript fully available?

Reviewer #1: Yes

Reviewer #2: No

Reviewer #3: Yes

4. Is the manuscript presented in an intelligible fashion and written in standard English?

Reviewer #1: Yes

Reviewer #2: Yes

Reviewer #3: No

5. Review Comments to the Author

Reviewer #1: Previous studies have identified risk factors for IgA nephropathy progression: eGFR and macrohematuria at the time of renal biopsy (Shu, D., Xu, F., Su, Z. et al. Risk factors of progressive IgA nephropathy which progress to end stage renal disease within ten years: a case–control study. BMC Nephrol 18, 11 (2017).

This study from Kensuke Joh et al evaluates the diagnostic approach for the treatment of IgA Nephropathy. The number of renal biopsies, the duration of follow-up and the quality of the statistical analysis permit the novelty of these useful and interesting conclusions:

Only tubular atrophy/interstitial fibrosis had an accelerative direct effect on renal function decline, while endocapillary hypercellularity and active crescents had an attenuating indirect effect via steroid treatment. This message is very important. A renal biopsy must be performed as soon as possible in order to initiate treatment for endocapillary hypercellularity and active crescent.

This study satisfies all the criteria to be accepted by PLOS One

I have only very few comments to this article: there are some acronyms in the abstract that are not identified and difficult the complete understanding of the message.

line 43 S and M

line 44 UPE

line 46 eGFR0c

Reviewer #2: The present study was a prospective clinical study for patients with IgAN among 44 kidney centers across Japan. By using the Oxford classification, the pathological variables were defined in the research and analysis. The authors applied structural equation modeling (SEM) to evaluate structural correlations associated with the change in eGFR, predicting the systemic path links between renal functional decline (RFD) and histological variables vis clinical or treatment variables. This is the only systematic research that proposes the correlation between histological, clinical, and treatment variables. Overall, it is an exciting study, and most of the conclusions may provide some information for IgAN patients using personalized medicine. However, I have some concerns about this study.

1. It is hard to catch the data and analysis process in the manuscript. A flow chart will help in this regard.

2. Figure 1 shows those variables for the SEM. The authors should outline the possible relationships between those variables but not just indicate the paths in the single image.

3. After removing non-significant paths is not clear. How to remove it? What is the difference without removal?

4. Can the authors simply the models? So complicated with the variables and sub-variables in this study. Making a table for results and conclusions will be much easier to follow.

5. The description for figure 2 is confusing.

Reviewer #3: This an paper dealing with finding of independent prognostic variables for renal functional decline in IgA nephropathy. Although the subject is of great interest, overall the article is not clear, with to many abbreviation within the text. Therefore, the manuscript is difficult to read. Some abbreviations are unnecessary – eg. UPE could be replaced with proteinuria (1 word), ST- steroids etc . Some of the abbreviations found in abstract must be spelled out completely on initial appearance in text (S,M,C, MEST…)

In table 1 The period of follow up overlaps with Treatment choice. Please clarify.

The authors should explain the raison for choosing different parameters like square root eGFR0 instead of eGFR; furthermore, it is not clear what does represent the values UPE0c, SReGFR0c and MAPc- why not MAP0c? …and the confusion continues with C0, C1 etc ….

The square root GFR could not be considered a predictor of SLOPE since GFR slopes depends on the eGFR.

The authors should also emphasize on the clinical values of their findings. A limitation section should be added as well.

6. PLOS authors have the option to publish the peer review history of their article (what does this mean?). If published, this will include your full peer review and any attached files.

Reviewer #1: **Yes: **Teresa Adragao

Reviewer #2: No

Reviewer #3: No

---

## [Author Response · Author response to Decision Letter 0]

1 Aug 2022

Dear Editor-in-Chief, PLOS ONE

Thank you for the thoughtful and constructive feedback you provided regarding our manuscript, entitled “Structural modeling for Oxford histological classifications of immunoglobulin A nephropathy.” We agree with you and have incorporated these suggestions throughout our paper.

With these changes to our final manuscript, we hereby resubmit our manuscript for a secondary evaluation. Thank you once again for your consideration of our paper.

Answers for the reviewer’s comments

Reviewer #1: Previous studies have identified risk factors for IgA nephropathy progression: eGFR and macrohematuria at the time of renal biopsy (Shu, D., Xu, F., Su, Z. et al. Risk factors of progressive IgA nephropathy which progress to end stage renal disease within ten years: a case–control study. BMC Nephrol 18, 11 (2017).

Thank you for introducing an interesting paper. For the present study, we have selected clinical variables according to the study by Barbour et al., who selected amount of proteinuria, eGFR, and MAP but not macrohematuria (JAMA Intern Med. 2019;179: 942-52). The information on macrohematuria was unfortunately not collected in the national-wide prospective study and therefore not used in the present study. 

This study from Kensuke Joh et al evaluates the diagnostic approach for the treatment of IgA Nephropathy. The number of renal biopsies, the duration of follow-up and the quality of the statistical analysis permit the novelty of these useful and interesting conclusions:

Only tubular atrophy/interstitial fibrosis had an accelerative direct effect on renal function decline, while endocapillary hypercellularity and active crescents had an attenuating indirect effect via steroid treatment. This message is very important. A renal biopsy must be performed as soon as possible in order to initiate treatment for endocapillary hypercellularity and active crescent.

This study satisfies all the criteria to be accepted by PLOS One

I have only very few comments to this article: there are some acronyms in the abstract that are not identified and difficult the complete understanding of the message.

Line 43 S and Line 44 M

line 44 UPE

line 46 eGFR0c

We have changed some acronyms to their expanded terms in the abstract section.

Reviewer #2: The present study was a prospective clinical study for patients with IgAN among 44 kidney centers across Japan. By using the Oxford classification, the pathological variables were defined in the research and analysis. The authors applied structural equation modeling (SEM) to evaluate structural correlations associated with the change in eGFR, predicting the systemic path links between renal functional decline (RFD) and histological variables vis clinical or treatment variables. This is the only systematic research that proposes the correlation between histological, clinical, and treatment variables. Overall, it is an exciting study, and most of the conclusions may provide some information for IgAN patients using personalized medicine. However, I have some concerns about this study.

1. It is hard to catch the data and analysis process in the manuscript. A flow chart will help in this regard.

Thank you for your valuable suggestion. We have created a flow chart and added the text as Supplementary Material 1 (Page 14; Line 262).

2. Figure 1 shows those variables for the SEM. The authors should outline the possible relationships between those variables but not just indicate the paths in the single image. After removing non-significant paths is not clear. How to remove it? What is the difference without removal?

The final relationship was illustrated in Fig. 2 corresponding with yellow as clinical variables and green as treatment variables, among which all significant relationships were connected by the arrows. Blue lines show a direct relationship with the slope. All histological variables showed additional indirect effects on SLOPE via clinical variables or ST (red arrows). An analyzing process was shown in a flow chart (Supplementary Material 1), indicating the results of linear regression analysis among nonsignificant histological, clinical, and treatment variables (see Step 3 in Supplementary Material 1). We selected the significant paths from the result of Step 3 by removing nonsignificant paths (P > 0.05). SEM was performed again to attest to the goodness of fit for an appropriately fitting model (see Step 4 in Supplementary Material 1).

2. Can the authors simply the models? So complicated with the variables and sub-variables in this study. Making a table for results and conclusions will be much easier to follow.

An analyzing process was shown in a flow chart (Supplementary Material 1). The final result was shown in Table 3, showing the direct and indirect effects of Oxford histological variables (M, E, S, T, and C) on changes in the estimated glomerular filtration rate (SLOPE). To introduce Table 3, direct and/or indirect effects of histological variables on SLOPE were calculated as total coefficients, which were the total of direct and indirect coefficients, where each indirect coefficient of M, E, S, T, or C in Table 3 indicated the integration of each coefficient of the clinical and treatment variables in Table 2 according to the paths shown in Fig. 2. 

The description for figure 2 is confusing.

Figure 2 showed an appropriately fitting structural modeling. Using this model, Table 3 was made as the conclusion showing direct and indirect effects of Oxford histological variables on changes in the estimated glomerular filtration rate (SLOPE). 

Reviewer #3: This paper dealing with finding of independent prognostic variables for renal functional decline in IgA nephropathy. Although the subject is of great interest, overall, the article is not clear, with too many abbreviations within the text. Therefore, the manuscript is difficult to read. Some abbreviations are unnecessary – eg. UPE could be replaced with proteinuria (1 word), ST- steroids etc. 

We agree that several abbreviations are found within the text; therefore, we should avoid some unnecessary abbreviations. However, each abbreviation has its original meaning, e.g., UPEc0 (centralized urine protein excretion (UPE), SReGFR0c centralized square root estimated glomerular filtration rate (eGFR), MAPc01: centralized mean arterial pressure (MAP) dichotomized with negative MAPc as MAPc0 and positive MAPc as MAPc1. ST not only means steroid therapy but also steroid therapy with/without tonsillectomy. Therefore, we believe that it is better to use abbreviations than to spell out on an initial appearance in the text. Please let us know if you still need the abbreviations to be defined at some places.

Some of the abbreviations found in abstract must be spelled out completely on initial appearance in text (S,M,C, MEST…)

In the abstract, we have spelled out the initial appearance of S and M. We used proteinuria instead of UPE in the text (Page 3; Lines 34–46).

In table 1 The period of follow up overlaps with Treatment choice. Please clarify.

In Table 1, the follow-up period was separately indicated from the treatment choice as ST and RASB.

The authors should explain the raison for choosing different parameters like square root eGFR0 instead of eGFR; furthermore, it is not clear what does represent the values UPE0c, SReGFR0c and MAPc- why not MAP0c? …and the confusion continues with C0, C1 etc ….

To obtain the goodness of fit for an appropriately fitting model, we changed the clinical variables respecting the variable normality, such as UPEc0 (centralized urine protein excretion (UPE), SReGFR0c centralized square root estimated glomerular filtration rate (eGFR), MAPc01: centralized mean arterial pressure (MAP) dichotomized with negative MAPc as MAPc0 and positive MAPc as MAPc1. The final choice of centralization (UPEc0), square root (centralized square root eGFR), or dichotomized MAPc01 was the best choice to obtain the most appropriate fitting model.

The square root GFR could not be considered a predictor of SLOPE since GFR slopes depends on the eGFR.

We believe that eGFR at biopsy was high, and subsequently, SLOPE declined strongly, whereas when it was low, there was a weaker decline. This is also true in normal kidneys. Therefore, according to our understanding, eGFR at the time of biopsy can influence SLOPE.

The authors should also emphasize on the clinical values of their findings. A limitation section should be added as well.

Thank you for your valuable comments. We added the following sentence in the conclusion (Page 21; Lines 390–393).

“We focused not only on direct effect but also on the indirect effect of histological variables on RFD, where direct effects of clinical variables on RFD, such as SReGFR0c, UPE0c, and ST, were influenced by histological variables.”

We added the following information to the limitation section (Page 21; Lines 398–401). 

“As a limitation, this prospective study using SEM could not predict the prognosis of renal function via Cox regression analysis because the goodness of fit could not be applied to the survival analysis model. Obtaining a more apparent eGFR slope needs longer follow-up.” 

Sincerely,

Kensuke Joh, MD, PhD

Department of Pathology, The Jikei University School of Medicine,

3-25-8 Nishi-Shinbashi, Minato-Ku, Tokyo, 105-8461, Japan

Tel: 81-3-3433-1111, Fax: 81-3-5472-0700

E-mail: johken@med.tohoku.ac.jp

---

## [Decision Letter · Decision Letter 1]

17 Aug 2022

Structural modeling for Oxford histological classification s of immunoglobulin A nephropathy

PONE-D-22-13403R1

Dear Dr. Joh,

We’re pleased to inform you that your manuscript has been judged scientifically suitable for publication and will be formally accepted for publication once it meets all outstanding technical requirements.

Kind regards,

Zhanjun Jia

Academic Editor

PLOS ONE

Additional Editor Comments (optional):

Reviewers' comments:

Reviewer's Responses to Questions

**Comments to the Author**

1. If the authors have adequately addressed your comments raised in a previous round of review and you feel that this manuscript is now acceptable for publication, you may indicate that here to bypass the “Comments to the Author” section, enter your conflict of interest statement in the “Confidential to Editor” section, and submit your "Accept" recommendation.

Reviewer #1: All comments have been addressed

Reviewer #2: All comments have been addressed

2. Is the manuscript technically sound, and do the data support the conclusions?

Reviewer #1: Yes

Reviewer #2: Yes

3. Has the statistical analysis been performed appropriately and rigorously? 

Reviewer #1: Yes

Reviewer #2: N/A

4. Have the authors made all data underlying the findings in their manuscript fully available?

Reviewer #1: Yes

Reviewer #2: Yes

5. Is the manuscript presented in an intelligible fashion and written in standard English?

Reviewer #1: Yes

Reviewer #2: Yes

6. Review Comments to the Author

Reviewer #1: This study from Kensuke Joh et al evaluates the diagnostic approach for the treatment of IgA Nephropathy. The number of renal biopsies, the duration of follow-up and the quality of the statistical analysis permit the novelty of useful and interesting conclusions. It is possible to change the decline of renal function based on the results of renal biopsy.

The final review of this article answers my previous comments

Reviewer #2: The authors have addressed all the questions, and the current version of the manuscript has dramatically improved and become more readable.

7. PLOS authors have the option to publish the peer review history of their article (what does this mean?). If published, this will include your full peer review and any attached files.

Reviewer #1: **Yes: **Teresa Adragao MD PhD

Reviewer #2: No

---

## [Editor Report · Acceptance letter]

31 Aug 2022

PONE-D-22-13403R1 

Structural modeling for Oxford histological classifications of immunoglobulin A nephropathy 

Dear Dr. Joh:

I'm pleased to inform you that your manuscript has been deemed suitable for publication in PLOS ONE. Congratulations! Your manuscript is now with our production department. 

Kind regards, 

on behalf of

Dr. Zhanjun Jia 

Academic Editor

PLOS ONE